# A Free-Standing Polymer Polypyrrole/Cellulose Composite Film via Spatial-Confined Interfacial Electrodeposition for Flexible Supercapacitors

**DOI:** 10.3390/ma16196476

**Published:** 2023-09-29

**Authors:** Sijie Wang, Wen Chen, Xinyue Huang, Xuezheng Chen, De Li, Feng Yu, Yong Chen

**Affiliations:** 1State Key Laboratory of Marine Resource Utilization in South China Sea, Hainan Provincial Key Laboratory of Research on Utilization of Si-Zr-Ti Resources, Hainan University, Haikou 570228, China; wwsj1207@163.com (S.W.); chenwen996@126.com (W.C.); huangxinyue122@163.com (X.H.); c18760162082@163.com (X.C.); lidenju@sina.com (D.L.); 2Guangdong Key Laboratory for Hydrogen Energy Technologies, School of Materials Science and Hydrogen Energy, Foshan University, Foshan 528000, China

**Keywords:** flexible surpercapacitor, polypyrrole, cellulose, flexibility

## Abstract

As a kind of energy storage device, a flexible supercapacitor has the characteristics of high capacity, fast charge/discharge rate, good stability, portability and softness. Conductive polymer polypyrrole (PPy) can be used as an electrode material for supercapacitors due to its environmental friendliness, simple synthesis process, good conductivity and potential for large-scale production. However, pristine PPy inevitably suffers from structural rupture due to repeated doping/de-doping during charge and discharge processes, which in turn impairs its cycle stability. In general, compounding with flexible substrates like soft carbon materials, cellulose or nylon fabric, is a good strategy to weaken the inner stress and restrain the structure pulverization of PPy. Herein, cellulose is utilized as a soft substrate to compound with PPy based on the electrochemical oxidation of polypyrrole. The interfacial electrodeposition method can successfully obtain a smooth, uniform and flexible PPy/cellulose composite film, which shows good conductivity. The assembled symmetric supercapacitor with PPy/cellulose film has an optimized specific capacitance of 256.1 mF cm^−2^, even after 10,000 cycles at a current density of 1 mA cm^−2^. Furthermore, there is no significant capacitance loss even after 180° bending of the device. This work provides a new means to prepare flexible, low-cost, environmentally friendly and high-performance electrode materials for energy conversion and storage systems.

## 1. Introduction

Exploring clean and renewable energy is an effective way to solve the energy crisis. Natural energy sources, such as wind, solar and tidal energy, are affected by geographical and natural conditions, so energy storage devices need to be developed. Usually, energy storage devices can be divided into electrostatic capacitors, chemical batteries, and supercapacitors (SCs) [1]. Compared with electrostatic capacitors and chemical batteries, supercapacitors exhibit a higher power density, a faster storage rate, and a longer service life [2,3,4,5]. Recently, portable wearable electronic devices have attracted widespread attention [6,7,8], such as foldable-screen mobile phones, implantable medical devices, microrobots [9,10] and even energy harvesting [11,12,13,14]. Among them, flexible supercapacitors (FSCs) are promising energy storage devices because of their good mechanical flexibility, good safety performance and low influence on performance under mechanical deformation [15,16,17,18,19]. At present, flexible supercapacitor electrodes have been developed in various carbon nanomaterials [20], metal oxides (TMOs) [21], conducting polymers (CPs) and their composites [22]. As one of the representative CPs, PPy is considered as one of the most promising materials due to its good electrical conductivity, easy synthesis, environmental friendliness, and low cost [23,24,25,26].

Generally speaking, flexible supercapacitor electrodes include self-supporting flexible electrodes and flexible substrate electrodes. Normally, a flexible substrate electrode can be easily prepared by mixing powder-shaped active materials with an organic binder and conductive carbon black, and then coating the mixture on flexible metal collectors like Cu foil, Al foil and Ni foams. However, the introduced binder and carbon black not only increase the mass of the electrode, but also weaken the flexibility of the electrode. Moreover, the low electrical conductivity and relatively poor stability of organic binders hinder the electron mobility inside the electrode and the ion diffusion on the electrode surface, resulting in poor electrochemical performance [27]. Furthermore, the metal collectors can be easily oxidized, leading to a decrease in conductivity. In addition, the deformation of metal collectors is irreversible after prolonged stretching, which usually leads to some cracks appearing in the electrode. Therefore, it is indispensable to design and prepare binder-free and self-supporting flexible supercapacitor electrodes.

Self-supporting electrodes are usually prepared via electrostatic spinning, chemical vapor deposition, vacuum filtration and electrodeposition [28,29,30,31]. Luo et al. [32] prepared a PPy/BP (black phosphorus) self-supporting film without adding any binder or conductive agent via a simple electrodeposition method, which can deliver high specific capacitance of 497.5 F g^−1^ and good cycling stability. However, the mechanical properties of this self-supporting electrode could not withstand behaviors like stretching and bending [33]. Combining self-supporting PPy film with flexible substrate materials including common plastics, textiles, graphene paper, cellulose paper, and other polymer substrates is a good strategy to improve the tensile and compressive properties of electrodes for supercapacitors [34]. However, the affinity between the flexible plastic substrate and the PPy film is weak, which may cause the active material to peel off from the substrate after repeated mechanical deformation, thus leading to decreased durability. 

In this work, a self-supporting flexible PPy/cellulose composite film was designed via spatial-confined interfacial electrodeposition to solve the abovementioned problems. Here, allyl chloride modified cellulose was chosen as the flexible substrate for PPy, owing to its advantages of abundant reserves, environmental friendliness, high mechanical properties, strong flexibility and low cost [35,36,37]. The introduction of allyl groups can not only reduce the crystallinity of cellulose but also improve its solubility, so as to ensure the ultraviolet-induced crosslinking process [38]. The obtained PPy/cellulose composite film through the interfacial electrodeposition method is smooth, uniform and flexible, as well as having good conductivity. By regulating the electrodeposition time, monolayer or multiple PPy films can be easily obtained and the electrochemical properties of Ppy/cellulose electrodes are significantly enhanced.

## 2. Experimental Methods

### 2.1. Materials

Allyl-modified cellulose (AC) was synthesized in our laboratory. Pyrrole (Py) was purchased from Macklin Co., Ltd. (Shanghai, China). Dimethyl sulfoxide (DMSO) was acquired from Aladdin Co., Ltd. (Fukuoka, Japan). Initiator 2959 (I2959) was bought from BASF. Oil of vitriol (H_2_SO_4_) was obtained from Xilong Chemical Co., Ltd. (Shantou, China). Sodium perchlorate (NaClO_4_) was purchased from AsahiKASEI (Shanghai, China). All reagents were of analytical grade and used without further purification. Distilled water was used throughout the experiment.

### 2.2. Preparation of AC Precursor Solution

AC was prepared according to our previous work [37]. First, 500 mg of AC was dissolved in 8 mL of DMSO solvent, and then 2.5 mg of the I2959 (0.5% in mass of AC) was added to the abovementioned solution. Finally, the AC precursor solution was obtained after constant stirring for 1 h at room temperature.

### 2.3. Synthesis of Ppy/Cellulose (F-PC) Film

The F-PC film was fabricated via the electrochemical deposition method. Firstly, fluorine-doped SnO_2_ conductive glass (FTO, 30 × 30 × 2.2 mm, 7 Ω) was ultrasonically cleaned in ethanol and deionized water to remove surface impurities before drying. Subsequently, 200 μL of AC solution was evenly spread on FTO before it was irradiated under ultraviolet (UV) light for 120 s to form a cellulose gel film. Meanwhile, the pyrrole/NaClO_4_ electrolyte (with a volume ration of 1:20) was prepared through adding pyrrole monomer into a 0.3 M NaClO_4_ aqueous solution. Then, electrochemical deposition was conducted at a potential of 0.6 V in the pyrrole/NaClO_4_ electrolyte with a three-electrode system on the electrochemical workstation (Biologic VSP-300, Hong Kong, China). The FTO/AC was used as the working electrode, while Pt foil and Ag/AgCl were used as the counter and reference electrodes, respectively. The products were rinsed with distilled water and peeled off from the FTO glass to obtain Ppy/cellulose films. The as-fabricated Ppy/cellulose films were marked as F-PC-1200, F-PC-2400 and F-PC-4800 according to the deposition time of 1200, 2400, and 4800 s, respectively. The pure Ppy film prepared via electrodeposition was recorded as F-P to be as the control group.

### 2.4. Material Characterizations

The surface and cross-sectional morphologies of the samples were observed via scanning electron microscopy (SEM Phenom ProX, Shanghai, China). Raman spectroscopy was carried out using a DXRxi apparatus (Thermo Fisher, Massachusetts, USA). Electronic conductivity was tested using a four-probe conductivity tester (ST2722-SZ, Suzhou, China).

### 2.5. Electrochemical Measurements

To assemble a supercapacitor coin cell, two slices of electrode with the same size, 10 mm in diameter, were pre-immersed in 1 M H_2_SO_4_ aqueous solution for 6 h, and then assembled directly with the carbon cloth (pre-soaked in 1 M H_2_SO_4_). The electrochemical measurement was carried out on an electrochemical workstation (Biologic VSP-300, Hong Kong, China) and a Battery Test System (CT2001A, Wuhan, China). Cyclic voltammetry (CV) was performed at different scan rates, and galvanostatic charge/discharge (GCD) with different current densities were characterized under the voltage range of 0–0.6 V. Electrochemical impedance spectroscopy (EIS) was also used to test at the frequency range of 10^5^–10^−2^ Hz. The areal capacitance (Cp) of the two-electrode system was calculated based on the geometric area of F-PC film.

## 3. Results and Discussion

Excellent flexible supercapacitor electrode materials should have good flexibility and electrochemical properties. At present, the conductive polymer Ppy electrodes are usually prepared using the traditional film rolling process or electrochemical deposition. The traditional film rolling process involves mixing the Ppy with conductive agents and binders, rolling the film, and then pressing the obtained film onto the nickel foam collector to prepare the electrode. This process is complicated to perform and has a high production cost. Moreover, the usage of a binder and conductive agent as well as a metal collector increases the weight of the electrode, which greatly reduces the active materials loading mass within the electrode. Furthermore, conductive polymer electrodes prepared via the rolling process suffer from uneven distribution and insufficient conductivity, which ultimately results in a short lifetime of the supercapacitor. By contrast, the electrochemical deposition method polymerizes pyrrole monomers in situ through regulating the current or voltage. The electropolymerization method has the advantages of good controllability, a short reaction time and environmental friendliness, and so it has been widely used in the synthesis of Ppy films and their applications in sensors, photoelectric devices and other fields. In electrochemical polymerization, pyrrole monomer and anions in the solution are adsorbed onto the electrode, the adsorbed pyrrole monomer is oxidized into cationic free radicals (Py^+^), and combines with anions (ClO_4_^−^) to form neutral ion pairs (Py^+^ClO^−^).
(H-N-H)_ad_-e^−^→(H-N-H)_ad_^+^
(H-N-H)_ad_^+^ + ClO_4_^−^→(H-N-H)_ad_^+^ClO_4_^−^

The two ion pairs couple and shed two anions and two protons to form a pyrrole dimer.
2(H-N-H)_ad_^+^ClO_4_^−^→H-N-N-H + 2A^−^ + 2H^+^

The reoxidation of this dimer produces a cationic radical, which combines with an anion to form an ion pair.
H-N-N-H + ClO_4_^−^e^−^→A^−^(H-N-H)_ad_^+^

The two ion–ion pairs are coupled again, and through repeated oxidative coupling, the pyrrole conjugated chain is gradually lengthened, and finally polypyrrole is formed. 

As shown in Figure 1a, the morphology of F-P films prepared via the conventional surface deposition method is displayed in Figure 1b,d. It can be seen that Ppy nanoparticles aggregate and stack on the FTO substrate to form the F-P film. The F-P film prepared by means of this surface electrodeposition process is brittle and uneven, and is difficult to peel off from the FTO substrate. By contrast, as demonstrated in Figure 1b, the F-PC composite self-supporting flexible film is prepared via the interfacial electrochemical deposition method. The morphology of F-PC can also be observed in Figure 1a,c. Interestingly, it can be directly found that the F-PC film possesses a quite different morphology from that of the F-P film. The F-PC film has a very smooth surface and good flexibility, meaning that it can be easily peeled off from FTO substrate. The significant morphology difference between F-P and F-PC may be ascribed to the spatial limitation, as displayed in Figure 1e,f. When PPy chains grow at the interface of FTO and cellulose gel, its growth space is confined at the interface of two phases (Figure 1e). However, when PPy chains grow at the FTO surface only, they tend to aggregate as particles in 3D space (Figure 1f). Therefore, this is an efficient strategy to obtain smooth, flexible and self-supporting PPy films by confining the polymerization space at the interface of FTO/cellulose phases. This interfacial electrodeposition method exhibits greater advantages compared with the traditional preparation methods for PPy electrodes. Firstly, it does not require the participation of any conductive agent and binder. Secondly, flexible self-supporting PPy films can be obtained easily, which avoids the complicated film rolling process. Thirdly, it is low cost and environmentally friendly. Fourthly, the F-PC film formed via interfacial electrodeposition is very smooth and uniform, which is more beneficial for its electronic conductivity than that of the F-P film derived from surface electrodeposition.

In order to characterize the conductivity of F-P and F-PC films, the Raman spectra and four-probe conductivity test are carried out. As shown in Figure 2, Raman tests are performed on F-P and F-PC films. It has been reported that the intensity ratio (*I*_1574_/*I*_1500_) of the C=C stretching vibration peak at 1574 cm^−1^ to the PPy skeleton vibration peak at 1500 cm^−1^ can respond to the relative conjugation length of PPy chains [39]. Generally, the value of *I*_1574_/*I*_1500_ reflects the conjugation length and degree of polymer polymerization, which further determines the electrical conductivity. As shown in Table 1, the *I*_1574_/*I*_1500_ values of F-P and F-PC films are 1.28 and 3.75, respectively, being positively correlated with the conductivities of F-P (33.97 S cm^−1^) and F-PC films (62.03 S cm^−1^). As a result, the homogeneous and smooth F-PC films obtained from interfacial deposition show a longer conjugation length and better electrical conductivity compared to those of F-P. It is worth noting that the good conductivity and stable conjugated structure of PPy films are two critical factors which determine the cycling life of supercapacitors.

In order to explore the microstructure of F-PC composite film, the two surfaces and cross-section of the F-PC film is observed using SEM. As shown in Figure 3a, the F-PC film shows a bilayer structure with the PPy film tightly attached to the cellulose film. As shown in Figure 3b,c, the morphology of F-PC at the PPy side presents a continuous and folded film microstructure, while the other side is a smoother cellulose film. In order to further determine the bilayer structure of F-PC films, we performed EDS analyses on both sides of the film. As we know, the PPy consists of C, N, O, and H elements, while the modified cellulose film consists of only C, H, and O elements. As shown in Figure 3(b1), elements C, N, and O are tested on the side with folds, so the folded side can be identified as the PPy side. As shown in Figure 3(c1), only C and O elements are analyzed on the smooth side, so this side can be identified as the modified cellulose side.

The microstructure of PPy/cellulose films with different deposition times was physically photographed, as presented in Appendix A, and it can be observed that the PPy film can totally cover the whole FTO substrate after 900 s. The morphology of F-PC films with different reaction times (1200 s, 2400 s and 4800 s) is shown in Figure 4, denoted as F-PC-1200, F-PC-2400, and F-PC-4800, respectively. Interestingly, the thickness and number of layers of PPy films are strongly dependent on the deposition time. After 1200 s, the PPy film adheres to the cellulose film, as shown in Figure 4a, whose thickness is about 5 μm. When the deposition time increases to 2400 s, the second layer of PPy film continues to grow at the new interface between FTO and the first layer of PPy film, resulting in a new thin PPy film with the thickness of about 700 nm, as shown in Figure 4b. When continuing to increase the deposition time to 4800 s, the third PPy thin film can be observed clearly, as in Figure 4c. As the deposition time increases, the PPy grows layer by layer in the form of a film. However, due to the smooth surface, the interface adhesive force between PPy films decreases, and as a result, the outermost PPy film is easily detached during repeated bending and folding processes.

The prepared F-PC films are subjected to bending and folding tests, as shown in Figure 5a. The flexibility of the films was improved significantly by laminating PPy with modified cellulose. The F-PC films are able to withstand bending, torsion, and folding without any fracture. On the contrary, the brittle F-P films could not bear the bending and folding treatment, as demonstrated in Figure 5b.

Furthermore, both F-P and F-PC films are assembled into asymmetrical coin cells to evaluate their electrochemical behavior. The capacitive performance of F-P-1200, F-PC-1200, F-PC-2400 and F-PC-4800 electrodes is measured via an electrochemical workstation and the Blue-Cell test system. As shown in Figure 6a, at a scan rate of 50 mV s^−1^, the CV curves of F-C and F-PC electrodes both exhibit a rectangular-like shape, indicating a fast reversible capacitive behavior. Obviously, the relatively equal integral areas of F-PC-1200 and F-P-1200 indicate that the presence of cellulose does not contribute much to the capacitance. Interestingly, with the deposition time increasing, the integral areas of F-PC-2400 and F-PC-4800 curves gradually increase and F-PC-4800 shows the largest curve area. This should be attributed to the increase in PPy with the increase in the deposition time, resulting in the mass increase in the amount of active material. Figure 6b delivers the Nyquist plots of F-P-1200, F-PC-1200, F-PC-2400 and F-PC-4800, in which F-P-1200 film shows the best interfacial compatibility due to the single component. For F-PC composite films, they all possess high interface impedance because of the introduction of inert cellulose components. Among them, F-PC-4800 has the lowest charge transfer resistance (R_ct_) and the highest ionic conductivity. This may be ascribed to the high PPy content in F-PC-4800 that can absorb abundant electrolyte. As can be seen from Figure 6c, at a current density of 1 mA cm^−2^, the GCD curves of all four electrode materials are approximately linear and almost triangular, with no significant IR drop, which is consistent with their low internal resistance. According to the rate performance of F-P-1200, F-PC-1200, F-PC-2400 and F-PC-4800 electrodes (at current densities of 1, 5, 10 and 20 mA cm^−2^, respectively) shown in Figure 6d, F-PC electrode materials show good capacitance performance at a high current density.

The cycle performance of F-P-1200, F-PC-1200, F-PC-2400 and F-PC-4800 electrodes under the current density of 1 mA cm ^−2^ is studied. It can be seen from Figure 6e that the areal capacitance of F-PC electrodes gradually increases with the increase in the deposition time. This increased initial capacity is ascribed to the aggregated PPy content after a long deposition process. The capacity retention of the above four samples after 10,000 cycles is 97.50%, 83.13%, 66.25% and 42.77%, respectively, showing decreased cycle stability as deposition time increases. The poor cycle performance of F-PC-4800 might be due to the poor interface adhesion within films, which blocks the transmission of electrons. However, F-PC-4800 still maintains the highest areal capacitance compared with other samples of about 250 mF cm^−2^, even after 10,000 cycles.

In order to verify the feasibility of using the F-PC films in flexible supercapacitors, a bending test on the F-PC electrode is carried out, in which the F-PC electrode material is repeatedly folded at 180° 100 times. CV curves and long-term cycle performance of the treated electrode clearly show the results of this bending test. As shown in Figure 7a, the CV curves show that the integral area of the folded F-PC electrode is almost unchanged, which illustrates that the electrode materials before and after folding 100 times exhibit the same capacitance behavior. The long-term cycle performance after folding is also tested at a current density of 1 mA cm^−2^. Figure 7b shows the comparison of results before and after folding. Compared with the original F-PC electrode, the initial capacity of the folded F-PC electrode does not change much. It can be verified from long-term cycle performance that the prepared F-PC electrode material can still maintain excellent performance under the condition of repeated folding. As a result, this F-PC film has great application potential in the direction of wearable devices.

To further illustrate the practicability of F-PC films as flexible electrodes, a luminescence test is conducted. Given that the working potential window of PPy film aqueous supercapacitors is 0.6 V, slightly lower relative to the potential in daily use, an appropriate method is employed by connecting multiple SSCs in series and in parallel to improve practicality. More intuitively, three symmetrical devices are assembled in series to reach the voltage of 1.8 V, as shown in Appendix A, which can successfully light a small bulb. Moreover, five symmetrical devices are assembled in series (achieving to the operating voltage window of 3 V) to successfully light up the lamp board.

## 4. Conclusions

In conclusion, F-PC composite self-supporting films are successfully designed via the interfacial electrodepositing method. PPy films can be spatially restricted to grow at the interface of FTO and cellulose phases, which endows these PPy films with smooth, flexible, homogenous surface and excellent conductivity. Significantly, F-PC-1200 film electrodes assembled into symmetric supercapacitors provide a capacitance of 136.33 mF cm^−2^, showing a retention rate of 83.13% after 10,000 cycles (1 mA cm^−2^). Moreover, these capacitors also show excellent electrochemical stability, even under extreme bending deformation conditions. These results suggest that the designed F-PC composite film shows great potential in wearable flexible devices.

## Data Availability

The data are reported within the article.

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
