# Peer review of "A Free-Standing Polymer Polypyrrole/Cellulose Composite Film via Spatial-Confined Interfacial Electrodeposition for Flexible Supercapacitors"

_materials, 2023, doi:10.3390/ma16196476_

Round 1
Reviewer 1 Report
The authors reported a novel method to obtain robust conductive polypyrrole membrane by deploying cellulose as a soft substrate. The flexible PPy/cellulose composite film as well as good conductive. The assembled symmetric supercapacitor base on PPy/cellulose film has optimized specific capacitance of 256.1 mF cm-2 even after 10000 cycles at a current density of 1 mA cm-1. Therefore, I would recommend publishing. However, the author must carefully review the abbreviations in their material. While simple spelling mistakes can still convey the author's intent to readers, if abbreviations from two different materials are mixed up, it could greatly affect the readers' interpretation. For instance, the authors claim PPy as F-PC and PPy/cellulose as F-P in scheme 1, but authors claim “Synthesis of PPy/ allyl-modified cellulose as F-PC film” (Line 106). Similar mistakes were also found in Figure 5.
Moreover, the characterization should be done after a long cycle test to demonstrate the stability of this material.
Reviewer 2 Report
The submitted work entitled “A free-standing PPy/cellulose composite film via spatial-confined interfacial deposition for flexible supercapacitor” by S. Wang was submitted to Materials/MDPI. Flexible electrode materials have attracted extensive attention in the applications of supercapacitors. The design and synthesis of well-performing electrodes with high mechanical flexibility are crucial. Conductive film electrodes are regarded as ideal flexible electrode materials based on the characteristics of being binder-free, ultrathin, and lightweight. Recently, in the literature, the most used flexible electrode materials are paper-like carbon-based materials and electronically conducting polymers. The authors have discussed polymer polypyrrole with cellulose composites for asymmetric supercapacitor applications.
The paper is written OK but the clarity of the mechanism involved in the polymer during the electrochemical process must be justified through appropriate equations must be detailed.
Further to this, my specific points that require some attention are given below.
· What is FTO? Explain
· Line 38 “Flexible supercapacitors” F – small letter.
· How the crosslinking improve energy storage?
· Has the amount of PPy /cellulose, and electrodeposition time been optimized?
· Manickam Minakshi et al have published key papers in polymers (including chitosan, alginate, etc.) that include and benchmark the supercapacitor performance.
· The scales are not visible in the SEM images (Figure 1).
· What is the role of these functional groups present in the modified cellulose?
· The D/G band intensity ratio (ID/IG) must be calculated.
· Figure 5c – voltage is with respect to reference electrode?
· The key paper on supercapacitors published recently in a similar area doi.org/10.1016/j.progsolidstchem.2023.100390 must be included and discussed.
· Please provide the redox reactions involved in the Figure 5 CV curves.
· Why the cycling stability for F-PC – 4800 is unusual at the first 2000 cycles?
· What is the surface area of the material?
· Do the different deposition times affect the morphology of electrode materials and further influence their electrochemical performance?
It is fairly OK.
Reviewer 3 Report
The paper is devoted for free-standing PPy/cellulose composite film for flexible application. The topic is generally interesting, however contain unexplained places (below) and need major revisions.
Fig. 1 should be more discussed in the paper text.
More comparison of obtained results and results already published in literature should be added in the paper parts 2 and 3.
Additional investigations of samples with experimental techniques like DSC, TGA and others could be very usefull.
English need minor revisions.
Conclusions should be rewritten in more informative way.
English need minor revisions.
Reviewer 4 Report
The manucript
"A free-standing PPy/cellulose composite film via spatial confined interfacial deposition for flexible supercapacitor" is an interesting research.
Some remarks:
- In Abstract and Introduction sections provide the aim and the novelty of this research.
Some experimental methods are required FTIR and XRD in order to have a complete characterization of materials.
There were made the stability tests?
In Figure 1 the scale bar is not visible. Same for Figures 2 and 3.
Some minor English corrections are required.
Round 2
Reviewer 2 Report
The authors have fairly addressed my queries raised in the earlier round of review. Therefore, in this reviewer's opinion, the revised version is suitable for publication.
Reviewer 3 Report
Authors make proper corrections according to reviewer remarks
and I suggest to publish the paper as it is.
Reviewer 4 Report
I agree with publication.